# Using Mutation-Analysis to Examine an LLM's Ability to Summarize Code

Lara Khatib*
University of Waterloo
Waterloo, Canada
lara.khatib@uwaterloo.ca

Michael Pu*
University of Waterloo
Waterloo, Canada
michael.pu@uwaterloo.ca

Bogdan Vasilescu
Carnegie Mellon University
Pittsburgh, USA
bogdanv@andrew.cmu.edu

Meiyappan Nagappan
University of Waterloo
Waterloo, Canada
mei.nagappan@uwaterloo.ca

## Abstract

As developers increasingly rely on LLM-generated code summaries for documentation, testing, and review, it is important to study whether these summaries accurately reflect what the program actually does. LLMs often produce confident descriptions of what the code looks like it should do (intent), while missing subtle edge cases or logic changes that define what it actually does (behavior). We present a mutation-based evaluation methodology that directly tests whether a summary truly matches the code's logic. Our approach generates a summary, injects a targeted mutation into the code, and checks if the LLM updates its summary to reflect the new behavior. We validate it through three experiments totalling 624 mutation–summary evaluations across 62 programs. First, on 12 controlled synthetic programs with 324 mutations varying in type (statement, value, decision) and location (beginning, middle, end). We find that, for GPT-4, summary accuracy decreases sharply with complexity from 76.5% for single functions to 17.3% for multi-threaded systems, while mutation type and location exhibit weaker effects. Second, testing 150 mutated samples on 50 human-written programs from the Less Basic Python Problems (LBPP) dataset confirms the same failure patterns persist as models often describe algorithmic intent rather than actual mutated behavior with a summary accuracy rate of 49.3%. Furthermore, while a comparison between GPT-4 and GPT-5.2 shows a substantial performance leap (from 49.3% to 85.3%) and an improved ability to identify mutations as "bugs", both models continue to struggle with distinguishing implementation details from standard algorithmic patterns. This work establishes mutation analysis as a systematic approach for assessing whether LLM-generated summaries reflect program behavior rather than superficial textual patterns, enabling direct comparison across models, prompts, and datasets.

*Both authors contributed equally to this research.

AIware '26, Montreal, QC, Canada
© 2026 Copyright held by the owner/author(s).
ACM ISBN 979-8-4007-2601-9/2026/07
https://doi.org/10.1145/3805760.3814892

## CCS Concepts

• **Software and its engineering**; • **Computing methodologies** → *Artificial intelligence*;

## Keywords

Code Summarization, Large Language Models, Mutation Analysis

**ACM Reference Format:**
Lara Khatib, Michael Pu, Bogdan Vasilescu, and Meiyappan Nagappan. 2026. Using Mutation-Analysis to Examine an LLM's Ability to Summarize Code. In *Proceedings of the 3rd ACM International Conference on AI-Powered Software (AIware '26), July 6–7, 2026, Montreal, QC, Canada.* ACM, New York, NY, USA, 10 pages. https://doi.org/10.1145/3805760.3814892

## 1 Introduction

While writing code may be the most visible part of software development, a significant portion of a developer's time is actually spent trying to understand code that already exists. [11, 23]. Whether it's reviewing a teammate's work, working with legacy systems, or joining a new team, developers constantly need to read and comprehend a piece of code. In fast-paced development environments, code summaries offer a simple and effective way to quickly understand what a piece of code does. A good summary can save time, cognitive effort, and help with routine coding tasks like debugging, reviewing, onboarding, or working with unfamiliar codebases. On the other hand, an inaccurate or incomplete summary can waste time, or lead developers to draw the wrong conclusions about how the code works, which may lead to errors and defects in the software.

The emergence of tools powered by large language models (LLMs) such as ChatGPT [27], and GitHub Copilot [8] has fundamentally reshaped how developers interact with code [24, 26, 29]. Among their many applications, code summarization has received growing attention, with developers increasingly relying on these systems to interpret unfamiliar code. While the output of these tools often appears confident, it's not always clear whether it reflects what the code actually does. Prior work has shown that even experienced developers can misinterpret code due to small but deceptive patterns known as "atoms of confusion" [6, 7, 9]. These are short snippets of code that look harmless but cause misunderstanding because of how they are written or where they appear. Just like certain patterns can lead humans to make mistakes, large language models may also misrepresent parts of the code leading to summaries that sound plausible but are inaccurate. This raises the

question of whether these summaries truly track the underlying behavior of the code or merely reflect surface level patterns.

One way to evaluate how well a human summarizes code is to ask them to explain it, then check whether that explanation reflect what the code actually does [13]. If a change is made to the code, then the explanation should change too. This paper adopts a similar approach to evaluate LLM-based code summarization by comparing the summaries a model generates before and after a behavior-changing modification is applied to the code. Our approach checks whether a summary updates when the code behavior changes. This motivates an evaluation that tests whether summaries are grounded in the provided code. Since LLMs are trained to predict likely text, their summaries may rely on familiar patterns and common implementations which can lead them to describe what the code appears intended to do rather than what it actually does after a subtle edit. To introduce these changes in a systematic manner, we apply *mutations* to transform the code. Mutations are a concept from software testing [2] where faults are introduced into correct programs to evaluate test suites. We use them as a way to introduce meaningful changes for evaluating code summaries.

Listing 1 shows an example of a mutation applied to a function. The model (GPT-4 in this case) summarizes both the original and mutated versions as "This method returns the smallest element in the heap" despite the fact that the mutated version returns the last element instead. Throughout the paper and in our replication package, we provide many more examples of cases where LLMs produces accurate summaries and where it fails to describe the actual behavior of the code. By comparing summaries across behavior-altering mutations, we can test whether a model is responding to the specific implementation details of the program or producing a description consistent with the intended pattern. Using this mutation-based evaluation, we observe a considerable number of cases where the model does not capture the introduced behavioral changes, even when they alter the intended outcome of the code.

```python
1  # Original Version
2  def get_min(self):
3      if not self.heap:
4          return None
5      return self.heap[0]
6
7  # Mutated Version
8  def get_min(self):
9      if not self.heap:
10         return None
11     return self.heap[-1]
```

**Listing 1: Snippet of original compared to snippet of val_e_2 from sample05. The value returned from the function is changed from the first element on line 5 to the last element on line 11.**

The primary contribution of this paper is the mutation-based evaluation methodology itself. We validate it through three complementary experiments that demonstrate its utility. First, we construct a controlled synthetic dataset designed to control for program size, mutation location, and structural complexity (spanning single-function to multi-threaded architectures). In this setting, we analyze performance as a function of program size and structural complexity, mutation type, and mutation location, allowing us to identify which dimensions most strongly affect whether a summary reflects

a change in the code's logic. Second, we evaluate the same methodology on a dataset of human-written programs to test whether the observed findings persist when the code is not model-generated. This addresses the concern that results from model-generated code may not transfer to code written by developers. Third, we repeat the evaluation across a newer generation model, enabling a direct comparison of whether newer models are more capable of tracking the code's logic under the same evaluation conditions. This work positions mutation analysis as a practical evaluation framework for comparing LLM-generated code summaries across models, datasets, and program characteristics.

## 2 Background and Related Work

Several recent studies have investigated the capabilities of LLMs in explaining and summarizing code [5, 13, 17, 25, 26, 32, 36]. Sun et al. [34] conducted a systematic evaluation of code summarization using LLMs, analyzing prompting techniques, model settings, and performance across different programming languages. Ahmed et al. [1] investigated whether adding structured semantic facts from code to the LLM prompts improves code summarization, finding that this approach enhances summarization performance. Existing benchmarks like CodeXGLUE [19] and HumanEvalExplain [25] were also developed to evaluate code summarization performance. Most prior work in this field has primarily relied on metrics like BLEU [28], ROUGE-L [15], METEOR [4], or BERTScore [40] to assess summary quality. However, these metrics have been shown to correlate poorly with human judgments [21, 31]. To address this, recent research has explored alternative approaches. Wu et al. [39] introduced CodeRPE, a framework that uses LLMs as evaluators of code summaries by assigning them different roles to provide different perspectives. Sun et al. [35] studied summarization strategies for higher-level code units and investigated the use of LLMs as evaluators. Their findings suggest LLMs can approximate human judgment but may exhibit a bias toward the summaries generated by itself. Similarly, Mastropaolo et al. [21] proposed SIDE, a metric that uses contrastive learning to assess how well a summary aligns with the semantics of the code independent of reference summaries. In contrast to prior work that relies on automated metrics or LLM-based evaluation, we assess summary correctness through human judgment. We also focus on whether LLM-generated summaries accurately capture the behavior and the intent of the code providing a more comprehensive evaluation.

There has been a growing interest in evaluating how well LLMs understand code, especially using mutation-based or metamorphic testing approaches. Ma et al. [20] introduced an evaluation method for dynamic behavior understanding of code using mutation analysis. Specifically, they performed an equivalent mutant detection test where they prompted the LLM to determine whether a given mutant was equivalent to the original code. Li et al. [14] also adapt this idea and propose Mutation-based Consistency Testing (MCT) where small semantic-altering changes are injected into code. They then provide the LLM with the code and its human-written description to test whether the LLM can correctly detect subtle inconsistencies between the code and its natural language summary. Similarly, Haroon et al. [10] applied semantic-preserving code mutations (SPMs) to faulty real-world programs and test whether LLMs can locate

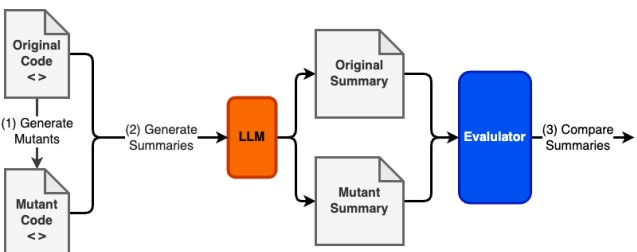

Figure 1: Overview of process.

the faults, essentially testing the code understanding ability of the LLM. Likewise, Wang et al. [38] explore using LLMs to generate mutants. They show that GPT-4 can produce syntactically valid mutations that are close to real-world bugs comparable to traditional mutation tools. Our work similarly adopts a mutation-testing perspective, but differs from all of the above in that it focuses on whether the model's summaries change when the underlying code is subtly modified. This shift lets us examine the model's internal consistency and sensitivity to semantic perturbations in code.

One could argue that our work is on code understanding and that we should look at it from that lens. However, based on feedback from the previous version of this paper, we recognize that LLMs rely on the Transformer architecture to statistically predict the most likely next token based on patterns learned from vast amounts of text [37]. They inherently are not understanding code. This is supported by research conducted at Anthropic [3] where they state that currently they are only able to see if models can think or not with prompts that are only 10 words long. Even with those and only in the case of poetry can they see that the model may think a few words ahead and not just the next word. In math problems they find that the models will not follow logical steps. All of this indicates that at the very least, we do not know if models can understand code or not. We can only see that it can summarize code. Therefore, we stick to code summarization as the topic for this paper

## 3 Research Questions

We validate our mutation-based evaluation methodology through three questions:

- **RQ1:** How do complexity, program size, mutation type, and mutation location influence the model's summarization performance?
- **RQ2:** Are the same limitations seen in human-written code?
- **RQ3:** How does summarization reliability change across model generations?

## 4 Experimental Process

We describe our experimental process, as outlined in Figure 1.

### (1) Generate Mutants

Given an original code sample, we first generate mutants by injecting changes into the original code. Each type of mutation targets a different aspect of the code and serves as a proxy for evaluating how well the model is able to capture each of these aspects. *(a)* ***Statement mutations*** remove, duplicate, or rearrange specific statements in the code. This results in statements being executed in

a different order, more than once, or not at all, causing changes in the program flow. *(b)* ***Value mutations*** modify parameters or constant values in the code and are intended to test the model's ability to reflect these changes in its summaries. *(c)* ***Decision mutations*** modify arithmetic or logic operators which affect the program's decision-making logic.

### (2) Generate Summaries

After the mutants have been generated, we produce summaries for each of the mutants as well as the original code. For this study, we define a *summary* as the natural language explanation of a code snippet's logic and behavior generated by an LLM. We produce these summaries by feeding the code into the model with the fixed prompt: ("Explain the following code snippet in plain English.") This prompt structure aligns with prior studies that concluded that the use of LLMs is beneficial [13, 26, 32]. To ensure reproducibility and prevent context leakage, we instantiate a fresh session for each sample and set the temperature to 0 to enforce deterministic output.

### (3) Compare Summaries

The final stage of our pipeline involves the *evaluator*, which compares the summary of the mutated code against the original to determine if the behavioral change was detected. We explicitly chose human evaluators over automated systems for two critical reasons. First, we rejected heuristic-based metrics such as BLEU [28]. While standard in Natural Language Processing, recent literature indicates these metrics correlate poorly with semantic correctness outside of machine translation tasks [30]. They are particularly ill-suited for code summarization, where functionally identical logic can be described in vastly different words, rendering n-gram overlap an unreliable proxy for accuracy. Second, we considered but ultimately rejected execution-based evaluation methods used in recent benchmarks [5, 16, 25]. These methods typically attempt to regenerate executable code from the summary and run unit tests. However, this approach lacks the precision required for our specific objective. Because our mutations are often subtle semantic perturbations (e.g., modifying a single comparator or return value), the noise introduced by the re-generation process could obscure whether the summary itself accurately captured the change. Instead, we compared the summaries manually, and checked whether we could identify the mutation by comparing the two summaries. If the two summaries were different and we were able to identify the change that was applied by comparing the two summaries, then the code sample was considered a *positive* result. If we were not able to identify the change from the two summaries, then the code sample was considered a *negative* result. This would mean that for two different pieces of code, the model generated the same summary and as a result, we conclude that it did not accurately summarize the code.

### 4.1 Positive

An example of a code sample that was classified as positive is `sample01/val/b2`. Listing 2 displays a snippet of the the original and mutated code samples. The summary for the original snippet says: *It first checks if the length of the list is greater than 1, because a list with only one element is already sorted.* and the summary for the mutated snippet says: *The function first checks if the length of*

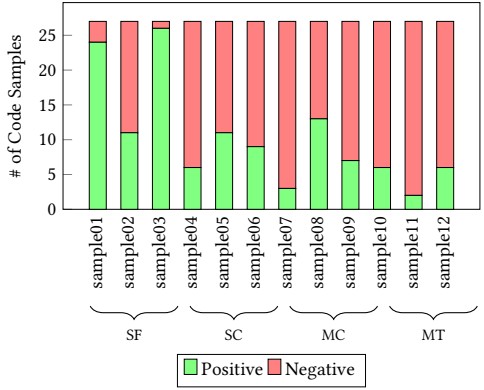

**Figure 2: Results of detecting mutations across the different code samples.**

```
1  # Original Version
2  def merge_sort(arr):
3      if len(arr) > 1:
4          mid = len(arr) // 2
5
6  # Mutated Version
7  def merge_sort(arr):
8      if len(arr) > 2:
9          mid = len(arr) // 2
```

**Listing 2: Snippet of original compared to snippet of val_b_2 from sample01. The '1' value on line 3 was changed to '2' on line 8.**

```
1  # Original Version
2  def merge_sort(arr):
3      if len(arr) > 1:
4          mid = len(arr) // 2
5
6  # Mutated Version
7  def merge_sort(arr):
8      if len(arr) > 1:
9          mid = len(arr) // 3
```

**Listing 3: Snippet of original compared to snippet of val_b_1 from sample01. The '2' value on line 4 was changed to '3' on line 9.**

*the array 'arr' is greater than 2.* This shows that the mutation was clearly identified in the summaries and that they are different. Other code samples that have been classified in this category also had their mutations surfaced in the summaries with a similar level of clarity. In some cases, the mutation is not only identified, but also recognized as an error from the original code. An example of this is sample01/val/b1, which is shown in Listing 3. Here, the summary for the mutated code says: *However, there is a mistake in the code: the array is being split into two halves using 'len(arr) // 3' instead of 'len(arr) // 2'.* From this, it appears that the model recognized the intended merge sort algorithm and was actually able to point out the mistake and what to do to fix it.

## 4.2 Negative

One way a sample is classified as a negative if the mutated summary still describes the original behaviour. This means that the summary is wrong and does not accurately describe the behaviour of the mutated code. An example of this is sample01/desc/b2, as shown

```
1  # Original Version
2  def merge_sort(arr):
3      if len(arr) > 1:
4          mid = len(arr) // 2
5
6  # Mutated Version
7  def merge_sort(arr):
8      if len(arr) < 1:
9          mid = len(arr) // 2
```

**Listing 4: Snippet of original compared to snippet of desc_b_2 from sample01. The '<' sign on line 3 was changed to '>' on line 8.**

```
1  # Original Version
2  def find(parent, i):
3      if parent[i] == i:
4          return i
5      return find(parent, parent[i])
6
7  # Mutated Version
8  def find(parent, i):
9      if parent[i] != i:
10         return i
11     return find(parent, parent[i])
```

**Listing 5: Snippet of original compared to snippet of desc_b_1 from sample02. The '==' comparator on line 3 was changed to '!=' on line 9.**

in Listing 4. The summary for the mutated code still says *It first checks if the length of the list is greater than 1.*, even though the mutation changes the behaviour of the code to check if the length of the list is less than one. sample02/desc/b1 shows that even a mutation applied to a well-known algorithm can go unrecognized. Listing 5 shows the implementation of the find operation for a union-find data structure in Kruskal's algorithm, which is a well-known algorithm for finding the minimum spanning tree of a graph. The mutation inverts the logic for when a node is determined to be a parent of itself, however, this goes unnoticed by the model. The mutated summary still says: *If 'i' is not its own parent (indicating that it is part of a larger set), the function recursively calls itself until it finds the root.* This is the behaviour of the original code, and is no longer correct for the mutated version. Another way a sample could be classified as a negative is if neither summaries explicitly described the mutated behaviour. That is, if the original and mutated summary both accurately describe the original and mutated code. Revisiting our earlier example of Kruskal's algorithm, we observe a case of this in sample02/stmt/b2, shown in Listing 6. Here, we removed the return value from the find operation when a node is a parent of itself. Python will implicitly return None in this case instead of the index of the node itself. This change is incompatible with the rest of the code and will result in an error, but this is not recognized by the model. In Listing 7 are relevant snippets from the original and mutated summaries. We can see that the return value is not encoded into either summary and as a result, the mutation cannot be detected from the summaries. The difference between this case and the previous examples lies in the reason the mutation is not detected. In this case, the summaries generated are not precise or detailed enough to encode the parts of the code that are relevant to the mutation. On the other hand, in the previous example, the summaries do encode the relevant parts of the code, but the wrong information is encoded.

```
1  # Original Version
2  def find(parent, i):
3      if parent[i] == i:
4          return i
5      return find(parent, parent[i])
6
7  # Mutated Version
8  def find(parent, i):
9      if parent[i] == i:
10         return
11     return find(parent, parent[i])
```

**Listing 6: Snippet of original compared to snippet of stmt_b_2 from sample02. 'return i' on line 4 was changed to 'return' on line 10.**

```
Snippet of Original Summary:
`find(parent, i)`: This is a helper function that implements the
     "find" operation of the Union-Find data structure. It
     takes an array `parent` and an element `i`, and it
     recursively finds the root (representative) of the set
     that `i` belongs to. If `i` is its own parent, it is the
     root; otherwise, the function follows the chain of parents
     until it finds the root.

Snippet of Mutated Summary:
`find(parent, i)`: This is a helper function that implements the
     "find" operation of the Union-Find algorithm. It takes an
     array `parent` and an element `i`, and it recursively
     finds the root (or representative) of the set that `i`
     belongs to. If `i` is its own parent, it is the root.
```

**Listing 7: Snippets of summaries of the original compared to of stmt_b_2 from sample02**

## 5 Results and Discussion

In this section, we will answer the three research questions using the results we collected and discuss the implications of these answers.

### 5.1 RQ1: How do complexity, program size, mutation type, and mutation location influence the model's summarization performance?

**Dataset Selection and Rationale.** We evaluate summarization using two datasets, we describe the second one in RQ2 and RQ3. The first one consists of controlled synthetic programs: We construct 12 Python programs spanning four structural complexity categories: single function (SF), single class (SC), multiple classes (MC), and multiple classes with multithreading (MT), with three programs per category. This dataset is designed to support controlled analysis as it allows us to vary program complexity and to place mutations in specific regions of the code (beginning, middle, end) while keeping other factors fixed. The programs were created using GPT-4 with manually crafted prompts and manual edits.

**Mutated Samples.** For each of the 12 controlled programs, we generate variants by varying mutations along two dimensions. The first is mutation type: statement, value, and decision. The second is mutation location: we place the mutation in the first, middle, or last third of the code. To reduce the risk that results are driven by a single edit, we create multiple distinct mutants for each (type, location) combination. This design yields 324 mutated samples in total and enables an analysis of how summarization behavior

changes with program complexity, mutation type, and where the change occurs in the code.

**Location of Mutation:** This factor looks at whether the location of a change within the code affects how likely an LLM is to reflect it in its summary. The motivation behind this is the primacy/recency effect in humans [33], where humans tend to remember best items that appear first, followed by those at the end, and weakest for those in the middle. Since LLMs also use attention mechanisms that assign varying levels of importance to different parts of an input sequence [37], it is possible that a similar phenomenon may occur. Additionally, Liu et al. showed that large language models tend to perform poorly at identifying relevant information in the middle of large contexts compared to the beginning and end [18].

**Choice of Model.** We used GPT-4 (gpt-4-1106-preview) for RQ1, which was state-of-the-art at the time of study design (mid-2024) and remains widely deployed in research and practice. While newer models have since been released, the labor-intensive nature of manual classification (324 mutations with dual annotation) necessitated focusing on a single model for depth. Rather than replacing this data with every new release, we use it to establish a concrete reference point. By documenting the limitations of this established model first, we can accurately measure the progress of the field when we compare it against the newer GPT-5.2 model in RQ3.

**Inter-Rater Reliability.** Two authors independently classified the 324 mutants. Inter-rater agreement was 96.6%, with Cohen's $\kappa = 0.928$, indicating almost perfect agreement. Disagreements (3.4%) were resolved through discussion, and the reconciled labels were used in all analyses.

**Effect of Program Size and Complexity.** A chi-square test revealed a significant association between program complexity and summarization accuracy, $\chi^2(3) = 69.04$, $p < 0.001$, with a large effect size (Cramér's $V = 0.462$). Accuracy decreased monotonically as complexity increased: 76.5% for single-function programs, 33.3% for single-class programs, 28.4% for multi-class systems, and 17.3% for multi-threaded systems. To better visualize the results along this dimension, we present Figure 3, which aggregates the results by complexity categories. As code becomes more complex, summarizing it requires capturing relationships across functions, classes, and execution paths. Program size exhibited a similarly strong effect. Positive cases had a median size of 60 Lines of Code (LOC), whereas negative cases had a median size of 116 LOC. A Mann–Whitney $U$ test confirmed that this difference was significant ($U = 5481$, $p < 0.001$). These results highlight a challenge for GPT-4 based code summarization, as real-world code is typically larger, more complex, and composed of multiple interconnected components.

**Effect of Mutation Type and Location** In contrast, mutation type did not exhibit a statistically significant association with accuracy, $\chi^2(2) = 1.95$, $p = 0.378$. Likewise, mutation location within the program showed no significant association with summarization performance, $\chi^2(2) = 0.23$, $p = 0.890$. For better visualization, we present Figure 4 and 5 which shows the results aggregated by the mutation type and mutation location, respectively, across the different complexity categories. Within the SF complexity category, positive rates are nearly identical across mutation types (74–78%), suggesting mutation type has little practical effect on short single-function programs. As complexity increases, differences by mutation type become more pronounced: in SC, decision

mutations show the lowest positive rate (19%) compared to statement (33%) and value (48%), and in MC performance drops most for decision and statement mutations (22% and 15%) while value mutations remain higher (48%). Across categories, value mutations are generally handled better than statement or decision mutations, except in the MT setting where value mutations degrade sharply. One plausible explanation is that in multithreaded code, the same value can be changed by different threads at different times. This adds complexity that GPT-4 may not be able to reason about effectively, given that there are multiple threads of execution and values changing simultaneously. Additionally, we found no clear correlation between the location of a mutation and GPT-4's summarization accuracy, suggesting that all parts of the code are treated with similar attention during summarization.

**Modes of Failure: Missing Details vs. Hallucinated Intent.** Manual analysis reveals two failure modes. In the first, the summary is too abstract it omits the mutated logic entirely, describing high-level intent that applies equally to original and mutated versions. This reflects a granularity problem where the summary fails to capture implementation details. In the second, the summary explicitly describes the original program's behavior rather than the mutated version. Here the model produces a description aligned with expected algorithmic patterns even when the mutation contradicts that pattern. The second failure mode is particularly revealing. Consider sample01/stmt/m1: a j+=1 line is removed from merge sort, but the summary still claims the index is incremented, a canonical step in merge sort widely available in training data. Similarly, in sample06/desc/b3, the hashmap load factor calculation is mutated from self.size / self.capacity to self.size * self.capacity, yet the summary still says "If the load factor (size divided by capacity) exceeds 0.7." GPT-4 appears to recognize the function as hashmap insertion and assumes standard internal behavior without verifying the actual implementation. We also observe that string literals can mislead the model. In sample07/val/e2, the list_customers function originally prints customer names with their wallet values. The mutated version prints the cart contents instead but retains the "Wallet: " string prefix. GPT-4's summary for the mutated version still claims it "prints out the list of customers and their wallet balances" despite wallet values no longer being printed. The hard-coded string literal appears to anchor the model's interpretation, causing it to describe what the string suggests rather than what the code outputs.

**Implications.** These findings suggest that GPT-4's summarization relies heavily on structural cues, naming conventions, and typical implementation patterns. When code follows canonical patterns, the model describes the expected algorithm. When mutations introduce deviations, even to well-known algorithms, the model often continues describing the pattern it expects rather than the code it reads. As architectural complexity increases, this tendency produces progressively more failures because the model must track relationships across multiple components.

## 5.2 RQ2: Are the same limitations seen in human-written code?

**Dataset Selection and Rationale.** To evaluate whether the summarization limitation observed in RQ1 generalize to human-written code, we selected the Less Basic Python Problems (LBPP) dataset

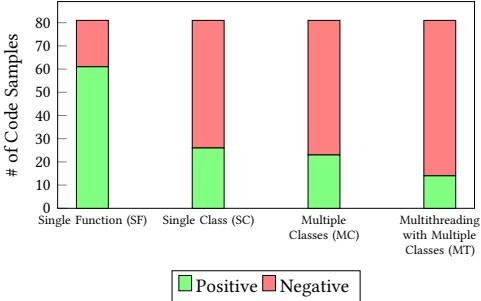

**Figure 3: Results across the complexity categories.**

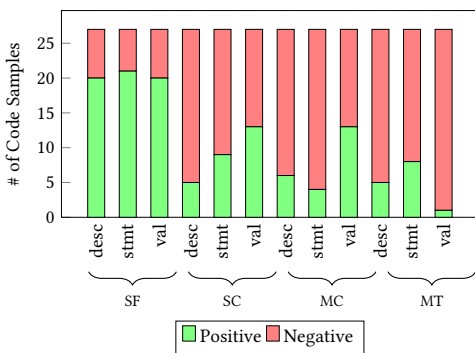

**Figure 4: Results across the different mutation types, grouped by complexity categories.**

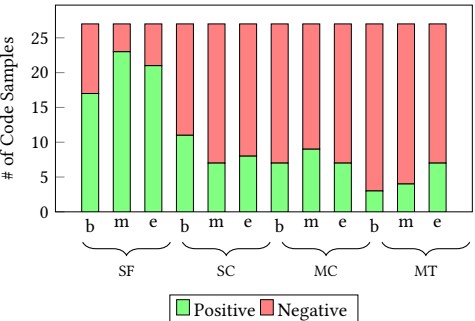

**Figure 5: Results across the different mutation locations, grouped by complexity categories.**

[22] as our evaluation corpus. LBPP is a collection of 161 programming problems specifically designed to be more challenging than benchmarks like HumanEval [5]. Importantly, LBPP was created through a process involving human annotators with competitive programming experience who were instructed to develop problems from scratch or from non-publicly available sources. The LBPP dataset offers several advantages for our evaluation purposes. First, it consists of human-written Python solutions that reflect real programming practices and patterns of human developers. Second, the problems are intentionally more difficult than other similar benchmarks. Third, LBPP was explicitly designed to be uncontaminated problems and solutions were verified not to overlap with existing training datasets, reducing the risk that GPT-4's performance might be affected by memorization. From the full LBPP dataset, we randomly selected 50 programs for our analysis. This subset size

was chosen to balance thoroughness with the manual evaluation effort required for our mutation-based methodology. While smaller than the controlled synthetic dataset used in RQ1 (12 programs with 324 total mutations), this sample size is appropriate given that our primary objective is to demonstrate that the mutation-based evaluation methodology can reveal similar limitations across different code sources. For each of the 50 programs, we applied three mutations (statement, decision, and value), yielding 150 mutated samples total for evaluation.

**Classification Process.** Given the high inter-rater agreement achieved in RQ1 (Cohen's $\kappa = 0.928$), the classification for RQ2 was performed by a single author who was part of the original annotation team. This approach maintains consistency with the established classification criteria while efficiently scaling the evaluation to include human-written code. We applied the same classification criteria ('Positive' vs. 'Negative') established in previous sections.

**Results.** Our manual classification results show that GPT-4 successfully detected 74 out of 150 mutations (49.3%). Breaking down by mutation type, we observe performance patterns consistent with RQ1. Value mutations achieved the highest detection rate at 58.0%, while statement and decision mutations showed similar, lower rates at 46.0% and 44.0% respectively. This ordering with value mutations being most reliably detected aligns with the trend observed in the controlled synthetic dataset, particularly in the lower complexity categories. The improved overall performance on LBPP (49.3%) compared to the synthetic dataset (38.9%) is primarily attributable to the structural characteristics of the programs. Most LBPP problems are function-level implementations, similar to the Single Function (SF) category in RQ1, which achieved a 76.5% detection rate. Because the LBPP dataset doesn't include complex multi-class or multi-threaded code, the average scores are naturally higher.

**Error Pattern Consistency.** Qualitative analysis of the misclassified cases in LBPP reveals error patterns strikingly similar to those documented in RQ1. We observed the same two failure modes: (1) summaries that are too abstract to capture the specific mutated logic, and (2) summaries that describe the intended or expected behavior of the original code rather than the actual behavior of the mutated version. For example, in LBPP task python/071, a statement mutation affecting node interleaving logic went undetected, with the mutated summary still describing the algorithm "as if it correctly appends alternating nodes," despite the mutation breaking this behavior. The most revealing are cases where GPT-4's summaries describe correct algorithmic patterns even when the code has been mutated to violate those patterns. In several graph algorithm implementations, mutations that changed program's logic were not reflected in the summaries, which continued to describe standard traversal or search procedures. This mirrors the findings from RQ1 where mutations to well-known algorithms (merge sort, heap operations, Kruskal's algorithm) were similarly missed when they deviated from canonical implementations.

**Implications** The results from RQ2 confirm that the limitations identified through mutation-based evaluation in the controlled synthetic setting are not artifacts of model-generated code. The same challenge persists in human-written code. The fact that similar failure patterns emerge across both datasets, particularly the tendency to summarize intent rather than actual behavior, suggests that these limitations are related to how current LLMs approach code summarization. This consistency validates our approach as a reliable tool for assessing summarization quality independent of how the code was produced, making it a practical evaluation framework for real-world applications where developers increasingly rely on LLM-generated code summaries to understand both human-written and AI-generated codebases.

## 5.3 RQ3: How does summarization reliability change across model generations?

**Motivation and Experimental Design** A key advantage of the mutation-based evaluation is its ability to enable model comparison. Unlike benchmark leaderboards that may reflect dataset contamination, our methodology applies identical mutations to the same code and measures whether different models detect the same behavioral changes. To demonstrate this capability, we compared GPT-4 (gpt-4-1106-preview, released November 2023) against GPT-5.2 (gpt-5.2, released January 2025) on the chosen set from the LBPP dataset [22] (50 programs, 150 mutations). Both models received identical prompts, identical code samples, and identical mutations, with classifications performed using the same criteria established in RQ2.

**Results.** The improvement found to be substantial as GPT-5.2 detects 85.3% of mutations compared to GPT-4's 49.3%. This 36.0 percentage point gain represents genuine progress in code summarization over 14 months of model development. For context, GPT-5.2's performance exceeds even the RQ1 single-function baseline (76.5%), suggesting that newer models are approaching reliability on simple functions. This improvement is consistent across all mutation categories as shown in Table 1, with the largest gains in decision mutations (+44.0pp) and substantial improvements in both value (+34.0pp) and statement (+30.0pp) mutations.

**What Changed?** Beyond the improvement, GPT-5.2 exhibits a different failure mode. When GPT-5.2 detects mutations, it frequently identifies them as errors. Analysis shows that in more than 48% of detected cases, GPT-5.2 includes explicit language like "will crash," "bug," "incorrect," or "should be." This pattern suggests GPT-5.2 is also recognizing when code is wrong. This makes sense for LBPP as competitive programming problems follow canonical algorithmic patterns, and mutations often violate these patterns in obvious ways. GPT-5.2 still missed 22/150 mutations (14.6%). Failures cluster in statement mutations (12 missed) compared to value mutations (4 missed), suggesting the model handles incorrect constant values better than control flow changes. Moreover, LBPP is mostly single-function code, so this is a best-case setting. Whether GPT-5.2 maintains performance on the complex multi-class and multi-threaded code remains an open question, we expect performance would degrade, though likely not to GPT-4's levels. In many cases, both models detect the mutation but still describe the behavior as if it were the intended behavior, rather than adjusting the explanation to reflect how the function's logic has actually changed. This could mean that the model's success could be driven by recognizing deviations from canonical patterns, not deep reasoning.

**Implications.** This comparison indicates that mutation analysis is a systematic way to measure progress in code summarization. By holding code, mutations, and prompts fixed across models separated by 14 months, we can isolate model capability as the only

variable. The data shows GPT-5.2 made substantial improvement under identical conditions. However, an 85.3% success rate on simple single-function code means roughly one in seven mutations still goes undetected, which is concerning for high-stakes systems. For practitioners building tools on top of LLM-generated summaries, this methodology provides a way to measure whether switching to a newer model actually improves summarization for their use case.

## 6 Threats to Validity

### 6.1 Internal Validity

**Choice of Dataset.** For RQ1, we used programs created with GPT-4 and manual edits. Since GPT-4 largely authored this code, it may find it easier to summarize than unfamiliar code. This could inflate RQ1 detection rates. However, we view this as establishing an upper bound: if GPT-4 struggles to summarize code it helped create, performance on truly unfamiliar code may be worse. RQ2 addresses this concern by testing human-written LBPP programs, where similar failure patterns emerged. Additionally, LBPP mainly contains short, single-function algorithm problems, which makes it a relatively simple setting compared to larger, multi-class or multi-threaded systems. Because of this, the results in RQ3 should be seen as performance in a best-case scenario. Real-world codebases are typically more interconnected and structurally complex, and performance may differ in those settings. We use LBPP to provide a controlled comparison, but the same mutation-based method can be applied to more complex repositories in future work.

**Classification Process.** Two authors independently classified the 324 RQ1 mutants with 96.6% agreement (Cohen's $\kappa = 0.928$). Disagreements were resolved through discussion. For RQ2 and RQ3, classification was performed by a single author from the original team, using the same criteria established during RQ1 and keeping the decision simple: whether the behavioral change could be identified from the two summaries. This approach balances thoroughness with the practical constraints of manual evaluation. We acknowledge that single-annotator classification carries some subjectivity. However, our core conclusion is that the mutation-based technique can be used to assess code summarization ability, and the evidence for this comes from the relative improvement from GPT-4 to GPT-5.2. Since summaries from both models were classified by the same annotator using the same criteria, any bias would affect both equally, so the relative comparison is not undermined. We provide all classifications in our replication package.

**Prompt Design.** We used the prompt "Explain the following code snippet in plain English," similar to prompts shown to be useful in prior work. While prompt engineering could potentially improve performance, our goal is to evaluate current summarization capabilities under realistic conditions rather than to optimize for maximum accuracy. The methodology supports testing alternative prompts, researchers can apply different prompting strategies and use mutation-based evaluation to measure their effectiveness.

### 6.2 External Validity

**Choice of Mutation Types.** The mutations we made to each code sample were constructed manually, following the guidelines for generating mutants from mutation testing [2]. Integration with automated mutation tools like MutPy [12] is feasible, provided

**Table 1: Mutation-detection accuracy by type for GPT-4 vs. GPT-5.2**

| Mutation Type | GPT-4 | GPT-5.2 | Improvement |
|---|---|---|---|
| Statement | 46.0% | 76% | +30.0pp |
| Decision | 44.0% | 88.0% | +44.0pp |
| Value | 58.0% | 92.0% | +34.0pp |
| Overall | 49.3% | 85.3% | +36.0pp |

the generated mutations produce behavioral changes detectable through summarization. Equivalent mutants, for example, would be unsuitable, but tools could generate candidates at scale and filter for those that alter program behavior. Although our mutations do not cover the full space of possible mutations, this is not a concern for our study. We simply use mutations as a systematic way to introduce controlled changes to the code. Our focus is on whether the model reflects these changes in its summaries, not on the mutations themselves. What matters for our analysis is that the mutation changes the behavior of the code in a meaningful way. That said, the point here is not to model "typical bugs," it is to make sure the code's behavior actually changes and then check whether the summary changes too. For a more realistic setup, the same evaluation can be run using mutations mined from commits, bug fixes, or domain-specific edits.

**Choice of Models.** RQ1 focuses on GPT-4, which was state-of-the-art at the time of study design and remains widely deployed in research and practice. We acknowledge that LLM development is rapid, making comprehensive multi-model evaluation challenging given the labor-intensive nature of manual classification. However, the same evaluation procedure can be repeated with different models. RQ3 demonstrates this by comparing GPT-4 to GPT-5.2 under identical conditions, revealing substantial improvement (49.3% → 85.3%) in mutation detection. While all current models share the transformer architecture [37], the magnitude of limitations varies across implementations. These findings indicate that mutation-based evaluation provides a systematic approach for comparing model performance across generations, though comprehensive evaluation of additional model families remains valuable future work.

## 7 Future Work

Our methodology opens several directions for future work, each of which builds directly on the framework introduced in this paper.

**Extending to Additional Models.** RQ3 illustrate the methodology on a single cross-generational comparison (GPT-4 vs. GPT-5.2). A natural extension is to apply the same procedure to additional model families, including code-specialized models such as Code Llama and DeepSeek-Coder, as well as other commercial models such as Claude and Gemini. This would clarify whether the limitations observed for GPT-4 are mitigated in newer or code-specialized models, or whether they reflect a more general limitation of current LLMs. Our preliminary validation on Gemini 2.5 Flash (one sample per complexity category, included in our replication package) showed the same failure patterns, suggesting that the limitations are not specific to the GPT family, but a systematic study is needed to confirm this.

**Extending to Complex Code and Real-World Repositories.** RQ3 was conducted on single-function code from LBPP. A natural extension is to apply the same procedure to the multi-class and multi-threaded programs from RQ1, and beyond that to large

real-world repositories with mutations mined from real commits, bug fixes, or domain-specific edits. This would clarify whether the sharp accuracy degradation observed for GPT-4 on complex code persists in newer models, and whether the same failure patterns appear in code with multiple files, dependencies, and architectural complexity.

**Prompt Variation Ablation.** Our experiments use a single fixed prompt that reflects how a developer without prompt-engineering experience would interact with an LLM. Prompt optimization is important future work, and mutation-based evaluation is itself a tool to test and compare different prompts for code summarization; by applying identical mutations under different prompting strategies and measuring detection rates, the methodology provides a way to evaluate whether a prompt change leads to a measurable improvement in summarization rather than only stylistic differences. This also helps separate model capability from prompt design as explanations for observed failures.

**Reducing Manual Effort with LLM-as-Judge.** The classification step in our methodology is currently manual. Because the task is binary and inter-rater agreement was high ($\kappa = 0.928$), it is plausible that an LLM judge could approximate human classification. A useful next step is to compare LLM-judge classifications against our human-annotated labels to determine whether automated classification is reliable enough to scale the methodology to larger datasets. One caveat is that using an LLM to evaluate summaries introduces a new question; whether disagreements stem from issues in the summary or in the judge. Validating LLM as a judge against human labels on a held-out subset before scaling would address this.

## 8 Conclusion

A convincing LLM-generated code summary is not enough. What matters is whether the summary actually follows the program's behavior. In this paper, we introduced a mutation-based evaluation methodology that answers this directly by applying controlled mutations that change what the code does, then measure whether the model's summary updates in a way that reflects the behavioral change. Across our experiments, we show that our mutation-based technique allows us to see how well a model can summarize. Summaries are most valuable and where failures are easiest to miss as it matters for downstream tasks documentation, test generation, and code review. If the summary reflects the expected intent of a familiar pattern instead of the program's actual behavior after a subtle edit, downstream tools and developers may inherit that mismatch. Our evaluation should move beyond surface similarity metrics, which have repeatedly been found to align poorly with human judgment in summarization settings. Our methodology addresses this by giving a repeatable stress test you can run on any repository. One can swap in mutations that match their domain, try alternative prompts, compare models, and quantify which configurations are genuinely more helpful. If we want to use LLM summaries as dependable engineering artifacts, they should react to changes in code's logic as reliably as a careful human reviewer would. Our mutation-based summarization evaluation provides a replicable path to measure and improve that standard.

The goal of this paper is to introduce the evaluation methodology, and we believe evaluation is a necessary precursor to improvement.

Our methodology helps measure code summarization performance and identifies cases where models describe intent over actual behavior. These findings are important for researchers building new techniques, as the methodology can be used to verify whether any attempted improvement, whether through prompting, fine-tuning, or other techniques, actually results in better summarization. Our mutation-based technique is also useful to IDE developers who build code summarization tools. Such developers face a choice when a newer LLM is released; should they stick with the current LLM or use the newer one that might be more expensive? Additionally, they may be editing harnesses in the tool to improve them. Our technique can be used to determine if such choices lead to a measurable improvement. They can choose the type and location of mutation, and the complexity and size of the code being mutated, to see when their changes are showing improvements. Our technique provides measurable metrics to compare one approach to another, making it valuable when IDE developers are trying to improve the code summarization abilities of their tools.

We outline concrete extensions to this framework, including evaluation on additional models, prompt variation, automated classification, and real-world repositories, in Section 7.

**Data Availability:** Our dataset and code are available at the following online repository: https://doi.org/10.5281/zenodo.15549625.

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
