# OpenReview forum: "Using Mutation-Analysis to Examine an LLM's Ability to Summarize Code"
_ACM.org/AIWare/2026/Conference — AIware 2026_

### Official Review · Reviewer_8neV · 2026-03-03

**Rating:** 2
**Confidence:** 4

**Review:**

Strengths：
1. The paper is well-written with a clear structure. The research idea of adapting mutation testing to evaluate code summarization is a valuable angle that provides a more behavioral-grounded assessment than traditional metrics like BLEU or ROUGE.
2. The experimental design in RQ1 is relatively thorough: the authors systematically control for multiple dimensions including mutation type, mutation location, and structural complexity, enabling fine-grained analysis of which factors most strongly affect summarization quality.
3. The authors employed dual independent annotation for RQ1 with high inter-rater agreement (Cohen's κ = 0.928), which strengthens the credibility of the classification results. The inclusion of a cross-generational model comparison (GPT-4 vs. GPT-5.2) further demonstrates the utility of the methodology as a benchmarking tool.

Weakness：
1. The experimental scale is small and the generalizability is limited. RQ1 uses only 12 synthetic programs, and RQ2/RQ3 use only 50 LBPP programs, all written in Python. The paper does not cover other mainstream languages such as Java or C/C++, nor does it involve real-world industrial codebases. Moreover, model selection is restricted to the OpenAI family (GPT-4 and GPT-5.2), lacking comparisons with Claude, Gemini, or open-source models (e.g., CodeLlama, DeepSeek-Coder), making it difficult to generalize the conclusions to LLMs as a whole.
2. The classification for RQ2 and RQ3 was performed by a single author, which is inconsistent with the dual-annotation standard established in RQ1. Although the paper justifies this by citing the high agreement in RQ1, single-annotator classification still carries non-negligible subjectivity bias, especially for borderline cases of partial detection on the LBPP dataset. The authors should at least perform dual annotation on a subset of RQ2/RQ3 to verify consistency.
3. The paper's most central finding, that summary accuracy degrades sharply with increasing complexity, was only validated on GPT-4. However, the GPT-5.2 comparison introduced in RQ3 was only conducted on LBPP (single-function level code) and was not replicated on the multi-class and multi-threaded complex code from RQ1. This is precisely the most valuable missing comparison: whether newer-generation models can mitigate the summarization degradation observed in complex code scenarios.
4. The evaluation adopts a simple binary classification (Positive/Negative), which is too coarse-grained. In practice, the model may partially detect a mutation (e.g., the summary mentions the relevant region but the description is imprecise). The lack of a finer-grained evaluation scale (such as a "Partially Detected" category or a severity score) results in the loss of important analytical information.
5. The prompt design is overly simplistic. The paper uses only a single fixed prompt ("Explain the following code snippet in plain English") and does not explore chain-of-thought prompting, line-by-line analysis instructions, or prompts that explicitly direct the model to focus on implementation details. Given that prompt engineering has a significant impact on LLM output quality, conclusions drawn from a single prompt may underestimate the model's actual capabilities.

Questions:
1. The 85.3% accuracy of GPT-5.2 on LBPP was achieved on single-function level code. Can the RQ3 experiment be extended to the multi-class/multi-threaded complex code from RQ1? This is critical for validating the paper's core conclusion.
2. The paper mentions that GPT-5.2 identifies mutations as "bugs" in over 48% of detected cases. Among these bug identifications, how many accurately describe the specific content of the mutation, and how many merely indicate in general terms that "the code has a problem"? Can the authors provide a more detailed categorization?
3. If a more instructive prompt were used (e.g., "Describe exactly what each line does, noting any unusual logic"), how much would the detection rate change? This is essential for distinguishing between "insufficient model capability" and "insufficient prompt guidance."
4. All mutations in this paper were manually constructed. Can the authors discuss the feasibility of combining this methodology with automated mutation tools (e.g., MutPy) to improve the scalability and practicality of the approach?

**Summary:**

This paper proposes a mutation-based evaluation methodology to test whether LLM-generated code summaries truly reflect the actual behavior of the program rather than merely describing superficial intent. The approach injects mutations (statement, value, and decision types) into the code and then compares the summaries generated by the LLM for the original and mutated versions to determine whether the model captures the behavioral change. The authors validate this methodology through three experiments totalling 624 mutation-summary evaluations across 62 programs: 324 mutations on 12 controlled synthetic programs, 150 mutations on 50 human-written programs from the LBPP dataset, and a cross-generational comparison between GPT-4 and GPT-5.2. Results show that summary accuracy decreases sharply with code complexity (from 76.5% for single functions to 17.3% for multi-threaded systems), and GPT-5.2 demonstrates substantial improvement over GPT-4 (49.3% → 85.3%) on single-function code.

---

> ### Author Response · Authors · 2026-03-17
>
> We thank you for your time and valuable feedback. Please find our responses to your questions below:
>
> **Q1: Extend RQ3 to multi-class/multi-threaded complex code?** This is a great suggestion and we appreciate the reviewer raising it. At the time of study design (mid-2024), GPT-4 was the leading model. Our experimental design follows a deliberate progression: RQ1 evaluates on a controlled synthetic dataset to isolate factors like complexity, mutation type, and location. RQ2 tests whether the same limitations generalize to human-written code. RQ3 compares GPT-4 and GPT-5.2 on the same human-written dataset to measure the effect of generational improvement. Extending each experiment to additional models and complexity levels would be highly valuable, but the manual classification process is extremely labor-intensive (RQ1 alone required hundreds of hours of dual annotation across 324 mutations) and the API costs for generating summaries across multiple models and datasets are also costly. The core contribution is the mutation-based evaluation methodology itself, not the performance of a specific model. Anyone can replicate the methodology to evaluate performance of different models, on different languages and datasets. We will make this framing clearer in the paper and emphasize that the decrease in summary accuracy is a specific finding to the GPT-4 model.
>
> **Q2: GPT-5.2 bug identification, specific vs. generic categorization?** Thank you for this insightful question. Our evaluation criterion is whether the summary accurately describes the behavior of the code not whether it identifies the mutation as a bug. When GPT-5.2 flags something as a "bug" what matters for our methodology is whether the summary still describes the original intended behavior (negative) or whether it reflects the actual behavior of the mutated code (positive). In real-world, developers are summarizing existing code that may already contain bugs, they are not injecting mutations. So in our case, the model recognizing that code deviates from expected behavior is itself a positive signal for its summarizing quality regardless of whether it labels it a "bug" specifically or describes the deviation generically. That said, this is an interesting angle for deeper analysis and as stated in the common reply something we can do in a journal extension.
>
> **Q3: Effect of more instructive prompts?** This is a great ablation study, but given the 8-page limit for AIware we do not have space to include it. Our prompt ("Explain the following code snippet in plain English") is a straightforward realistic prompt a developer would use, and is aligned with prior work (Leinonen et al. 2023, Nam et al. 2023). We see prompt optimization as important future work and will discuss it in the paper. Additionally, our proposed mutation-based evaluation provides a method for measuring the effectiveness of different prompts for code summarization.
>
> **Q4: Feasibility of automated mutation tools?** Our manually constructed mutations follow standard mutation testing guidelines (Andrews et al. 2006) so integration with tools like MutPy is feasible. The key requirement is that mutations produce meaningful behavioral changes detectable through summarization. Not all tool-generated mutations will meet this, for example equivalent mutants would be unsuitable. However, mutation tools could generate candidate mutations at scale which can then be filtered. We will add this to the paper.
>
> **Regarding the other concerns raised:**
>
> **Single annotator for RQ2/RQ3:** We appreciate this concern and we agree that dual annotation is the stronger standard. It was labor-intensive, RQ1's 324 samples required hundreds of hours of dual annotation, and the goal of RQ2 and RQ3 was primarily to validate that the mutation-based methodology works on human-written code and across model generations, rather than to make strong claims about one model's performance. Given the near-perfect agreement in RQ1 (κ=0.928, 96.6%) on the same binary task, we felt single-annotator classification by a member of the original team was reasonable. That said, we recognize it is a limitation and we are happy to dual-annotate a subset of RQ2/RQ3 and report inter-rater agreement to address this concern.
>
> **Binary classification:** In practice, when a developer reads an LLM-generated summary for review, debugging, or onboarding, the summary either communicates the actual behavior or it does not. A summary that mentions the relevant region but mischaracterizes the behavior, for example, in sample02/desc_b_1, GPT-4 detects parent[i] != i but still describes the original find logic, is functionally a miss as a developer relying on it would still draw the wrong conclusion. We do observe these partial-detection cases in our qualitative analysis and discuss them in the paper with examples.

---

> > ### Comment · Reviewer_8neV · 2026-03-18
> >
> > Thank you for the detailed response. However, my main concern regarding the single-annotator classification in RQ2/RQ3 remains unresolved at the current time. As this directly affects the credibility of the core results, I keep my score.

---

> > > ### Author Response · Authors · 2026-03-19
> > >
> > > Thank you for your reply. We would like to clarify one point about the single-annotator classification. While we agree that it affects credibility, this concern is more true when the raw results of the classification are what is used for the conclusion. In our case, our core conclusion is that our mutation-based technique can be used to assess code summarization ability. We will make that clear if accepted. The evidence for our core conclusion comes from the fact that there is a relative improvement from GPT4 to GPT5.2. Summaries from both models were done by the same annotator. Hence, credibility issues are minimized since we are only looking at relative improvement over absolute values.

---

### Official Review · Reviewer_UnSD · 2026-03-08

**Rating:** 3
**Confidence:** 4

**Review:**

Strengths

* A mutation-based evaluation design for code summarization.
* Relatively comprehensive evaluation across synthetic and real-world datasets.
* Open-source implementation.

Weaknesses

* The motivation is not sufficiently justified.
* Heavy reliance on manual classification limits scalability.
* Limited discussion on how the findings can improve LLM code summarization.

Comments for the authors

1. This paper proposes a new approach to assess the ability of LLMs in code summarization by evaluating whether they can capture and interpret the changes between original code and mutated code. However, the motivation for this evaluation setup is somewhat unclear. Code summarization and code change understanding are related but not necessarily equivalent abilities, and the paper does not clearly justify why performance on the latter should be regarded as a meaningful indicator of the former. In particular, if LLMs are primarily trained for standard code summarization tasks rather than change identification, it is unsurprising that they may struggle in this setting. This paper does not clearly explain how stronger change-capturing ability would translate into improved code summarization performance. I therefore encourage the authors to further clarify the motivation and validity of the proposed evaluation perspective.
2. The approach relies heavily on manual effort, which limits its practicality and scalability in real-world evaluation settings. More importantly, this makes the advantage of the proposed framework somewhat unclear. If evaluators still need to manually inspect the summaries for both the original code and the mutated code, it is not obvious what efficiency or methodological benefit this approach offers over a more direct evaluation strategy. For example, one could simply expand the dataset and manually review all examples in a conventional way. Such a setup appears more straightforward and may even be more appropriate than the current process. If the authors would like to maintain this evaluation paradigm, it may be worthwhile to explore whether an LLM-as-a-judge strategy could reduce the manual efforts and improve the practicality of the approach.
3. The paper also provides limited insight into how the findings could help improve the code summarization capabilities of LLMs. Although the paper does include some implications, they are not presented from the perspective of model improvement. Since hallucination and response deviation are common issues in LLMs, detecting inconsistencies should not be the final goal. Understanding how such findings can be used to improve the models is equally, if not more, important. One possible direction would be to investigate whether the ability to capture changes between an original code snippet and its mutated version can actually improve code summarization performance.
4. The evaluation is limited to GPT-4 and GPT-5.2, which raises concerns about representativeness. Although these are strong commercial LLMs, they are not specialized code models for code summarization. Evaluating additional code-oriented models would strengthen the paper and improve the clarity of the conclusions.

**Summary:**

This paper introduces a mutation-based evaluation methodology that directly tests whether an LLM-generated code summary truly matches the code’s logic. Specifically, the approach generates a summary, injects a targeted mutation into the code, and checks if the LLM updates its summary to reflect the new behavior. The authors validated the proposed approach through three experiments totaling 624 mutation–summary evaluations across 62 programs. The experimental results show that summary accuracy declines significantly as program complexity increases, while mutation type and location have relatively limited impact. Experiments on the LBPP dataset further confirm that models often summarize algorithmic intent instead of actual mutated behavior. Overall, the paper argues that mutation analysis offers a systematic, replicable, and practical way to assess and improve the reliability of LLM-generated code summaries.

---

> ### Author Response · Authors · 2026-03-18
>
> We thank you for the thoughtful and detailed review. We address each concern below:
>
> **Concern 1: Motivation code summarization vs. code change understanding** We appreciate this concern and want to clarify a key aspect of our methodology. We are not asking the model to detect changes or compare two versions. The model never sees both versions of the code. It independently summarizes two different programs in completely separate, fresh sessions with no shared context (we instantiate a new session for each sample and set temperature to 0).
>
> The reasoning is, if a summary accurately describes what a program does, then two programs that do different things should receive different summaries. Consider Listing 1 in our paper: the original get_min returns self.heap[0] (the first element), and the mutated version returns self.heap[-1] (the last element). GPT-4 summarizes both as "returns the smallest element in the heap." The model was never asked to compare anything, it just produced a factually incorrect summary for the mutated code. We use the behavior altering mutations to reveal summarization inaccuracies. This mirrors how we would evaluate a human, if we change what the code does and the person still gives the same explanation, their explanation was wrong for at least one version. We also test a necessary condition of correct summarization, a model that produces identical summaries for behaviorally different programs cannot be accurately summarizing at least one of them.
>
> **Concern 2: Heavy reliance on manual classification** We chose human annotation because the subject of our study is LLM-generated summaries. If we were to use an LLM as a judge to evaluate LLM-generated summaries, we would not be able to determine whether a disagreement stems from an issue in the summary being evaluated or an issue in the judge itself. Human annotation avoids this problem. That said, it is a binary task and we do see high inter-rater agreement (κ=0.928, 96.6%), so it would be worth looking at LLM-as-judge and comparing it against the human annotations to establish whether it can be used to automate the method. This is a great ablation that we would love to include, but we do not currently have space within the 8-page AIware limit.
>
> **Concern 3: Limited insight into improving code summarization** We appreciate the reviewer raising this important point. The goal of this paper is to introduce the evaluation methodology and we believe evaluation is a necessary precursor to improvement. Our methodology helps us measure the code summarization performance as it identifies intent over actual behavior. These findings are important for researchers building new techniques as it can be used to verify whether any attempted improvement, whether through prompting, fine-tuning, or other new techniques, actually results in better summarization.
>
> **Concern 4: Evaluation limited to GPT-4 and GPT-5.2** That’s a great point! Evaluating code-oriented models is an interesting angle, it would be worth seeing how models specifically trained for code compare under identical conditions. We want to clarify that the core contribution of our paper is the mutation-based evaluation methodology, not the performance of a specific model. Anyone can apply the same procedure to any model of their choice. RQ3 demonstrates this by comparing two models under identical conditions. When we report findings about GPT-4 or GPT-5.2, those are specific to those models. We also validated on Gemini 2.5 Flash (one sample per complexity category), which showed the same failure patterns. Results are in our replication package.

---

> > ### Comment · Reviewer_UnSD · 2026-03-19
> >
> > I thank the authors for their response, which fully addresses my concerns regarding the motivation. However, it remains unclear exactly how the proposed method can help improve the performance of LLMs. I would be willing to raise my score if the authors can provide further insight into the potential applications of their method for enhancing LLM capabilities.

---

> > > ### Author Response · Authors · 2026-03-19
> > >
> > > Thank you for the reply! We can add further insights into potential applications for improving LLM summarization capabilities, like the following, in the conclusion section:
> > >
> > > Our mutation-based technique is useful to IDE developers who build code summarization tools. Such developers have a choice to make when a newer LLM is released - should we stick with the current LLM or use the newer one that might be more expensive? Additionally, they may be editing harnesses in the tool to improve them. Our technique can be used to determine if such choices lead to a measurable improvement or not. They can choose the type and location of mutation, and the complexity and size of the code being mutated, to see when their changes are showing improvements. Our technique can give measurable metrics to compare one approach to another. Hence, our technique becomes invaluable when IDE developers are trying to improve the code summarization abilities of their tools.

---

### Official Review · Reviewer_Kavf · 2026-03-10

**Rating:** 3
**Confidence:** 4

**Review:**

**Strengths**
=============

*   The paper addresses an interesting topic, whether LLM-generated code summaries truly reflect program behavior.

*   The mutation evaluation is novel and provides a systematic way to test whether summaries track behavioral changes in code.

*   The study is comprehensive and combines synthetic programs, human-written code, and cross-model comparison, providing multiple perspectives on summarization reliability.

*   The paper identifies key failure modes, particularly the tendency of LLMs to summarize algorithmic intent rather than actual implementation details.

*   The dataset and code are publicly available, supporting reproducibility.


**Limitations**
---------------

*   The evaluation focuses mainly on GPT-4 and GPT-5.2, leaving out other widely used models.

*   The synthetic dataset used in the first experiment contains programs partially generated with GPT-4, which may bias results.

*   The LBPP dataset primarily contains short, single-function programs, limiting evaluation on large-scale real-world systems.

*   While mutations change program behavior, they may not fully represent real-world software bugs or developer modifications.

*   The study uses a single prompt for summary generation, which may not reflect real usage scenarios where prompts vary.


**Detailed Comments**
=====================

*   The evaluation considers only two LLM models. While GPT-4 and GPT-5.2 models are widely used, the study does not include other popular code-oriented models such as Claude, Code Llama, or DeepSeek. Including additional models would help determine whether the observed failure patterns are specific to these models or represent broader limitations of LLM-based code summarization.

*   The synthetic dataset used in the first experiment contains programs generated using GPT-4 and manually edited. Because the same model was partially responsible for generating these programs, it may find them easier to summarize than unfamiliar code. This could introduce bias and potentially inflate the observed summarization performance in the first experiment.

*   The LBPP dataset used in the second experiment mainly contains short, single-function programs. Such programs do not capture the complexity of real-world software systems that involve multiple files, classes, dependencies, and architectural components. As a result, the reported performance may not reflect how LLMs behave when summarizing large-scale production codebases.

*   The mutation strategy used in the evaluation introduces controlled behavioral changes, but these mutations may not fully represent realistic software defects or developer edits. Real-world code changes often involve multiple interacting modifications, refactoring, or contextual changes across multiple files. Consequently, the mutation-based evaluation may capture only a subset of the challenges present in real software evolution.

*   The experiments use a single fixed prompt for generating summaries. In practice, developers interact with LLMs using a variety of prompts, including requests for documentation, comments, or explanations of specific behaviors. The reliance on a single prompt limits the understanding of how prompt design may influence summarization accuracy and mutation detection


**Questions for the Authors**
=============================

**1-** Why were only GPT-4 and GPT-5.2 selected for evaluation? Would the results differ significantly for other models such as Claude, Code Llama, or DeepSeek?

**2-** Have you considered evaluating the mutation-based methodology on large real-world repositories rather than short benchmark programs?

**3-** Did you evaluate whether different prompt designs could improve the ability of LLMs to detect behavioral changes in code?

**Summary:**

This paper proposes a mutation-based methodology to evaluate the reliability of LLM-generated code summaries. The approach introduces behavior-changing mutations into code snippets and examines whether the model updates its summary to reflect the modified behavior. The authors conduct 624 mutation–summary evaluations across synthetic programs and human-written code, analyzing how factors such as program complexity, mutation type, and model generation affect summarization accuracy. The results show that LLMs often describe expected algorithmic intent rather than actual implementation behavior, and that summarization accuracy decreases significantly as program complexity increases.

---

> ### Author Response · Authors · 2026-03-18
>
> We thank you for the positive and encouraging review. We address your questions below:
>
> **Q1: Why only GPT-4 and GPT-5.2?** GPT-4 was the leading model at the time of study design (mid-2024), and the manual classification process is very labor-intensive, RQ1 alone took hundreds of hours of dual annotation across 324 mutations. We prioritized a thorough evaluation of a single model over a broader but less in-depth comparison with multiple models. We did validate on Gemini 2.5 Flash (one sample per complexity category) and saw the same failure patterns. Those results are in our replication package. The core contribution is our evaluation methodology and anyone can repeat it with Claude, DeepSeek, Code Llama, or any model of their choice. RQ3 already shows this by comparing GPT-4 and GPT-5.2 under identical conditions.
>
> **Q2: Large real-world repositories?** Thank you for raising this. There is nothing in the methodology that restricts it to short programs, it works on any codebase. We started with controlled programs in RQ1 to isolate specific factors (complexity, mutation type, location) and then validated on human-written code in RQ2. Real-world repositories introduce additional variables (multiple files, dependencies, architectural patterns) that would make it harder to control for other factors and would also require the evaluators to have a deep understanding of the code behavior before they can judge the summary.
>
> **Q3: Different prompt designs?** Our prompt ("Explain the following code snippet in plain English") is a realistic, straightforward prompt aligned with prior work (Leinonen et al. 2023, Nam et al. 2023), and reflects how a developer who does not know prompt engineering would interact with an LLM to understand code. We acknowledge this as a limitation, and one thing we think is interesting is that mutation-based evaluation could actually be used as a way to test and compare different prompts for code summarization. We could not fit a prompt ablation in the current 8-page paper, but we think it is a compelling direction.
>
> **Regarding the limitations raised:**
>
> **Synthetic dataset bias:** Thank you for raising this, it is a fair concern.  Our experimental design was deliberately scoped to programs generated by GPT-4 in order to control for style, complexity, and potential confounding variables. While this design decision restricts external validity, we believe our findings represent a lower bound on performance. If GPT-4 struggles to accurately summarize code it has generated (and which is likely in-distribution), then it may perform even less reliably on human-written code (which could contain idiosyncrasies or structures that are out-of-distribution). RQ2 validates this by testing whether the same results generalize to human-written code
>
> **Mutations vs. real-world bugs:** This is an interesting point. Our mutations are not meant to model real-world bugs, we use them as a controlled mechanism to change what the code does and then check whether the summary reflects that change. The point is not to simulate typical defects but to test whether the model is summarizing the actual behavior of the code or falling back on pattern recognition. That said, the same methodology could be applied using mutations mined from real commits or bug fixes which would be an interesting extension.

---

### Author Response · Authors · 2026-03-18
**General Comment**

We thank all three reviewers for their feedback and the time they put into reviewing our work. The suggestions have been extremely helpful in identifying areas where we can strengthen the paper.

The reviewers have raised great ideas for extending the work. Reviewer 8neV suggests extending RQ3 to multi-class/multi-threaded code (whether newer-generation models can mitigate the summarization degradation observed in complex code scenarios) and conducting a prompt variation ablation. Reviewer UnSD suggests exploring LLM-as-judge for automated evaluation and investigating how findings can directly improve model capabilities. Reviewer Kavf suggests evaluating on large real-world repositories and testing additional models and different prompts.

We agree that these are all valuable directions. However, given the 8-page limit for AIware and the labor-intensive nature of our manual annotation process (RQ1 alone required hundreds of hours of dual annotation across 324 mutations), we were not able to include all of these in the current submission. The goal of this paper is to introduce the mutation-based evaluation methodology as a systematic approach for assessing code summarization quality. We see this as a necessary first step, and the extensions suggested by the reviewers are next steps that build directly on it.

We will revise the paper to more clearly frame the contribution as the methodology itself, and highlight these directions as future work. We have fit as much as we can within the scope of an AIware paper, but we would be happy to incorporate the suggested extensions in a journal extension of this work. We appreciate the reviewers helping us see the broader potential here.